# A Learning Error Analysis for Structured Prediction with Approximate Inference

**Yuanbin Wu[1, 2], Man Lan[1, 2], Shiliang Sun[1], Qi Zhang[3], Xuanjing Huang[3]**
[1]School of Computer Science and Software Engineering, East China Normal University
[2]Shanghai Key Laboratory of Multidimensional Information Processing
[3]School of Computer Science, Fudan University
{ybwu, mlan, slsun}@cs.ecnu.edu.cn, {qz, xjhuang}@fudan.edu.cn

## Abstract

In this work, we try to understand the differences between exact and approximate inference algorithms in structured prediction. We compare the estimation and approximation error of both underestimate (e.g., greedy search) and overestimate (e.g., linear relaxation of integer programming) models. The result shows that, from the perspective of learning errors, performances of approximate inference could be as good as exact inference. The error analyses also suggest a new margin for existing learning algorithms. Empirical evaluations on text classification, sequential labelling and dependency parsing witness the success of approximate inference and the benefit of the proposed margin.

## 1   Introduction

Given an input $x \in \mathcal{X}$, structured prediction is the task of recovering a structure $y = h(x) \in \mathcal{Y}$, where $\mathcal{Y}$ is a set of combinatorial objects such as sequences (sequential labelling) and trees (syntactic parsing). Usually, the computation of $h(x)$ needs an inference (decoding) procedure to find an optimal $y$:

$$h(x) = \arg\max_{y \in \mathcal{Y}} \text{score}(x, y).$$

Solving the "$\arg\max$" operation is essential for training and testing structured prediction models, and it is also one of the most time-consuming parts due to its combinatorial natural. In practice, the inference problem often reduces to combinatorial optimization or integer programming problems, which are intractable in many cases. In order to accelerate models, faster approximate inference methods are usually applied. Examples include underestimation algorithms which output structures with suboptimal scores (e.g., greedy search, max-product belief propagation), and overestimation algorithms which output structures in a larger output space (e.g., linear relaxation of integer programming). Understanding the trade-offs between computational efficiency and statistical performance is important for designing effective structured prediction models [Chandrasekaran and Jordan, 2013].

Prior work [Kulesza and Pereira, 2007] shows that approximate inference may not be sufficient for learning a good statistical model, even with rigorous approximation guarantees. However, the successful application of various approximate inference algorithms motivates a deeper exploration of the topic. For example, the recent work [Globerson et al., 2015] shows that an approximate inference can achieve optimal results on grid graphs. In this work, instead of focusing on specific models and algorithms, we try to analyze general estimation and approximation errors for structured prediction with approximate inference.

Recall that given a hypothesis space $\mathcal{H}$, a learning algorithm $\mathcal{A}$ receives a set of training samples $S = \{(x_i, y_i)\}_{i=1}^{m}$ which are i.i.d. according to a distribution $\mathcal{D}$ on the space $\mathcal{X} \times \mathcal{Y}$, and returns a

hypothesis $\mathcal{A}(S) \in \mathcal{H}$. Let $e(h) = \mathbf{E}_{\mathcal{D}}l(y, h(x))$ be the risk of a hypothesis $h$ on $\mathcal{X} \times \mathcal{Y}$ ($l$ is a loss function), and $h^* = \arg\min_{h \in \mathcal{H}} e(h)$. Applying algorithm $\mathcal{A}$ will suffer two types of error:

$$e(\mathcal{A}(S)) = \underbrace{e(h^*)}_{\text{approximation}} + \underbrace{e(\mathcal{A}(S)) - e(h^*)}_{\text{estimation}}$$

The *estimation error* measures how close $\mathcal{A}(S)$ is to the best possible $h^*$; the *approximation error* measures whether $\mathcal{H}$ is suitable for $\mathcal{D}$, which only depends on the hypothesis space. Our main theoretical results are:

- For the estimation error, we show that, comparing with exact inference, overestimate inference always has larger estimation error, while underestimate inference can probably have smaller error. The results are based on the PAC-Bayes framework [McAllester, 2007] for structured prediction models.

- For the approximation error, we find that the errors of underestimate and exact inference are not comparable. On the other side, overestimate inference algorithms have a smaller approximation error than exact inference.

The results may explain the success of exact inference: it makes a good balance between the two errors. They also suggest that the learning performances of approximate inference can still be improved. Our contributions on empirical algorithms are two-fold.

First, following the PAC-Bayes error bounds, we propose to use a new margin (Definition 3) when working with approximate algorithms. It introduces a model parameter which can be tuned for different inference algorithms. We investigate three widely used structured prediction models with the new margin (structural SVM, structured perceptron and online passive-aggressive algorithm).

Second, we evaluate the algorithms on three NLP tasks: multi-class text classification (a special case of structured prediction), sequential labelling (chunking, POS tagging, word segmentation) and high-order non-projective dependency parsing. Results show that the proposed algorithms can benefit each structured prediction task.

## 2   Related Work

The first learning error analysis of structured prediction was given in [Collins, 2001]. The bounds depend on the number of candidate outputs of samples, which grow exponentially with the size of a sample. To tighten the result, Taskar et al. [2003] provided an improved covering number argument, where the dependency on the output space size is replaced by the $l_2$ norm of feature vectors, and London et al. [2013] showed that when the data exhibits weak dependence within each structure (collective stability), the bound's dependency on structure size could be improved. A concise analysis based on the PAC-Bayes framework was given in [McAllester, 2007]. It enjoys the advantages of Taskar et al.'s bound and has a simpler derivation. Besides the structured hinge loss, the PAC-Bayes framework was also applied to derive generalization bounds (and consistent results) for ramp and probit surrogate loss functions [McAllester and Keshet, 2011], and loss functions based on Gibbs decoders [Honorio and Jaakkola, 2016]. Recently, Cortes et al. [2016] proposed a new hypothesis space complexity measurement (factor graph complexity) by extending the Rademacher complexity, and they can get tighter bounds than [Taskar et al., 2003].

For approximate inference algorithms, theoretical results have been given for different learning scenarios, such as the cutting plane algorithm of structured SVMs [Finley and Joachims, 2008, Wang and Shawe-Taylor, 2009], subgradient descent [Martins et al., 2009], approximate inference via dual loss [Meshi et al., 2010], pseudo-max approach [Sontag et al., 2010], local learning with decomposed substructures [Samdani and Roth, 2012], perceptron [Huang et al., 2012], and amortized inference [Kundu et al., 2013, Chang et al., 2015]. Different from previous works, we try to give a general analysis of approximate inference algorithms which is independent of specific learning algorithms.

The concept of algorithmically separable is defined in [Kulesza and Pereira, 2007], it showed that without understanding combinations of learning and inference, the learning model could fail. Two recent works gave theoretical analyses on approximate inference showing that they could also obtain

promising performances: Globerson et al. [2015] showed that for a generative 2D grid models, a two-step approximate inference algorithm achieves optimal learning error. Meshi et al. [2016] showed that approximation based on LP relaxations are often tight in practice.

The PAC-Bayes approach was initiated by [McAllester, 1999]. Variants of the theory include Seeger's bound [Seeger, 2002], Catoni's bound [Catoni, 2007] and the works [Langford and Shawe-Taylor, 2002, Germain et al., 2009] on linear classifiers.

## 3 Learning Error Analyses

We will focus on structured prediction with linear discriminant functions. Define exact inference

$$h(x, w) = \arg\max_{y \in \mathcal{Y}} w^\mathsf{T} \Phi(x, y),$$

where $\Phi(x, y) \in \mathbf{R}^d$ is the feature vector, and $w$ is the parameter vector in $\mathbf{R}^d$. We consider two types of approximate inference algorithms, namely *underestimate approximation* and *overestimate approximation* [Finley and Joachims, 2008] [1].

**Definition 1.** Given a $w$, $h^-(x, w)$ is an underestimate approximation of $h(x, w)$ on a sample $x$ if

$$\rho w^\mathsf{T} \Phi(x, y^*) \leq w^\mathsf{T} \Phi(x, y^-) \leq w^\mathsf{T} \Phi(x, y^*)$$

for some $\rho > 0$, where $y^* = h(x, w), y^- = h^-(x, w) \in \mathcal{Y}$. Similarly, $h^+(x, w)$ is an overestimate approximation of $h(x, w)$ on sample $x$ if

$$w^\mathsf{T} \Phi(x, y^*) \leq w^\mathsf{T} \Phi(x, y^+) \leq \rho w^\mathsf{T} \Phi(x, y^*)$$

for some $\rho > 0$, where $y^+ = h^+(x, w) \in \bar{\mathcal{Y}}$ and $\mathcal{Y} \subseteq \bar{\mathcal{Y}}$.

Let $\mathcal{H}, \mathcal{H}^-, \mathcal{H}^+$ be hypothesis spaces containing $h$, $h^-$ and $h^+$ respectively: $\mathcal{H} = \{h(\cdot, w) | w \in \mathbf{R}^d\}$, $\mathcal{H}^- = \{h^-(\cdot, w) | \forall x \in \mathcal{X}, h^-(\cdot, w) \text{ is an underestimation}\}$, and $\mathcal{H}^+ = \{h^+(\cdot, w) | \forall x \in \mathcal{X}, h^+(\cdot, w) \text{ is an overestimation}\}$. Let $l(y, \hat{y}) \in [0, 1]$ be a structured loss function on $\mathcal{Y} \times \mathcal{Y}$ and $\mathrm{I}(\cdot)$ be a 0-1 valued function which equals 1 if the argument is true, 0 otherwise.

### 3.1 Estimation Error

Our analysis of the estimation error for approximate inference is based on the PAC-Bayes results for exact inference [McAllester, 2007]. PAC-Bayes is a framework for analyzing hypothesis $h(\cdot, w)$ with stochastic parameters: given an input $x$, first randomly select a parameter $w'$ according to some distribution $Q(w'|w)$, and then make a prediction using $h(x, w')$. Define

$$L(Q, \mathcal{D}, h(\cdot, w)) = \mathbf{E}_{\mathcal{D}, Q(w'|w)} l(y, h(x, w')), \quad L(Q, S, h(\cdot, w)) = \frac{1}{m} \sum_{i=1}^m \mathbf{E}_{Q(w'|w)} l(y_i, h(x_i, w')).$$

Given some prior distribution $P(w)$ on the model parameter $w$, the following PAC-Bayes Theorem [McAllester, 2003] gives an estimation error bound of $h(x, w)$.

**Lemma 2** (PAC-Bayes Theorem). *Given a $w$, for any distribution $\mathcal{D}$ over $\mathcal{X} \times \mathcal{Y}$, loss function $l(y, \hat{y}) \in [0, 1]$, prior distribution $P(w)$ over $w$, and $\delta \in [0, 1]$, we have with probability at least $1 - \delta$ (over the sample set S), the following holds for all posterior distribution $Q(w'|w)$:*

$$L(Q, \mathcal{D}, h(\cdot, w)) \leq L(Q, S, h(\cdot, w)) + \sqrt{\frac{D_{\mathrm{KL}}(Q \| P) + \ln \frac{m}{\delta}}{2(m - 1)}},$$

*where $D_{\mathrm{KL}}(Q \| P)$ is the KL divergence between $Q$ and $P$.*

**Definition 3.** For $\rho > 0$, we extend the definition of margin as $m_\rho(x, y, \hat{y}, w) \triangleq w^\mathsf{T}\Delta_\rho(x, y, \hat{y})$, where $\Delta_\rho(x, y, \hat{y}) \triangleq \rho\Phi(x, y) - \Phi(x, \hat{y})$.

Clearly, $m_\rho(x, y^*, y^-, w) \leq 0$ for underestimation, and $m_\rho(x, y^*, y^+, w) \geq 0$ for overestimation.

The following theorem gives an analysis of the estimation error for approximate inference. The proof (in the supplementary) is based on Theorem 2 of [McAllester, 2007], with emphasis on the approximation rate $\rho$.

**Theorem 4.** *For a training set $S = \{(x_i, y_i)\}_{i=1}^m$, assume $h'(x_i, w)$ is a $\rho_i$-approximation of $h(x_i, w)$ on $x_i$ for all $w$. Denote $\rho = \max_i \rho_i$ and $M_i = \max_y \|\Phi(x_i, y)\|_1$. Then, for any $\mathcal{D}$, $l(y, \hat{y}) \in [0, 1]$ and $\delta \in [0, 1]$, with probability at least $1 - \delta$, the following upper bound holds.*

$$L(Q, \mathcal{D}, h'(\cdot, w)) \leq \mathcal{L}(w, S) + \frac{\|w\|^2}{m} + \sqrt{\frac{(1+\rho)^2\|w\|^2 \ln\frac{2m\lambda_S}{\|w\|^2} + \ln\frac{m}{\delta}}{2(m-1)}}, \qquad (1)$$

$$\mathcal{L}(w, S) = \begin{cases} \frac{1}{m}\sum_{i=1}^m \max_y l(y_i, y)\mathrm{I}(m_{\rho_i}(x_i, y_i^*, y, w) \leq M_i) & \text{if } h'(\cdot, w) \in \mathcal{H}^- \\ \frac{1}{m}\sum_{i=1}^m \max_y l(y_i, y)\mathrm{I}(m_{\rho_i}(x_i, y_i^*, y, w) \geq -M_i) & \text{if } h'(\cdot, w) \in \mathcal{H}^+ \end{cases}$$

*where $y_i^* = h(x_i, w)$, $Q(w'|w)$ is Gaussian with identity covariance matrix and mean $(1 + \rho)\sqrt{2\ln\frac{2m\lambda_S}{\|w\|^2}}w$, $\lambda_S$ is the maximum number of non-zero features among samples in $S$: $\lambda_S = \max_{i,y}\|\Phi(x_i, y)\|_0$.*

We compare the bound in Theorem 4 for two hypotheses $h_1, h_2$ with approximation rate $\rho_{1,i}, \rho_{2,i}$ on sample $x_i$. Without loss of generality, we assume $w^\mathsf{T}\Phi(x_i, y_i^*) > 0$ and $\rho_{1,i} > \rho_{2,i}$.

In the case of underestimation, since $\{y|m_{\rho_{1,i}}(x_i, y_i^*, y, w) \leq M_i\} \subseteq \{y|m_{\rho_{2,i}}(x_i, y_i^*, y, w) \leq M_i\}$, $\mathcal{L}(w, S)$ of $h_1$ is smaller than that of $h_2$, but $h_1$ has a larger square root term. Thus, it is possible that underestimate approximation has a less estimation error than the exact inference. On the other hand, for overestimation, both $\mathcal{L}(w, S)$ and the square root term of $h_1$ are larger than those of $h_2$. It means that the more overestimation an inference algorithm makes, the larger estimation error it may suffer.

Theorem 4 requires that $h'(\cdot, w)$ attains approximation rate $\rho_i$ on $x_i$ for all possible $w$. This assumption could be restrictive for including many approximate inference algorithms. We will try to relax the requirement of Theorem 4 using the following measurement on stability of inference algorithms.

**Definition 5.** $h(x, w)$ is $\tau$-stable on a sample $x$ with respect to a norm $\|\cdot\|$ if for any $w'$

$$\frac{|w^\mathsf{T}\Phi(x, y) - w'^\mathsf{T}\Phi(x, y')|}{|w^\mathsf{T}\Phi(x, y)|} \leq \tau\frac{\|w - w'\|}{\|w\|},$$

where $y = h(x, w), y' = h(x, w')$.

**Theorem 6.** *Assume that $h'(x_i, w)$ is a $\rho_i$-approximation of $h(x_i, w)$ on the sample $x_i$, and $h'(\cdot, w)$ is $\tau$-stable on $S$ with respect to $\|\cdot\|_\infty$. Then with the same symbols in Theorem 4, $L(Q, \mathcal{D}, h'(\cdot, w))$ is upper bounded by*

$$\mathcal{L}(w, S) + \frac{\|w\|^2}{m} + \sqrt{\frac{(1+2\rho+\tau)^2\|w\|^2 \ln\frac{2m\lambda_S}{\|w\|^2} + \ln\frac{m}{\delta}}{2(m-1)}}.$$

Note that we still need to consider all possible $w'$ according to the definition of $\tau$. However, upper bounds of $\tau$ could be derived for some approximate inference algorithms. As an example, we discuss the linear programming relaxation (LP-relaxation) of integer linear programming, which covers a broad range of approximate inference algorithms. The $\tau$-stability of LP-relaxation can be obtained from perturbation theory of linear programming [Renegar, 1994, 1995].

**Theorem 7** (Proposition 2.5 of [Renegar, 1995])**.** *For a feasible linear programming*

$$\max. \ w^\mathsf{T}z \quad s.t. \ Az \leq b, \ z \geq 0,$$

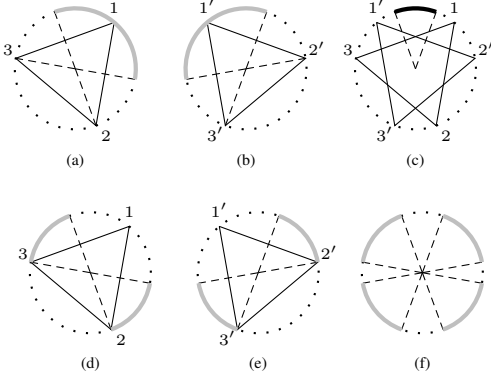

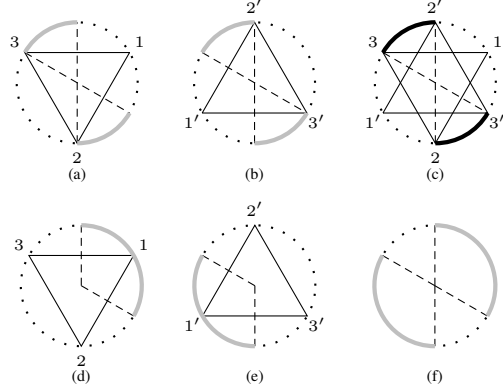

Figure 1: An example of exact inference with less approximation error than underestimate inference (i.e., $e(h) < e(h^-)$)

Figure 2: An example of underestimate inference with less approximation error than exact inference (i.e., $e(h^-) < e(h)$).

*let $\hat{z}$, $\hat{z}'$ be solutions of the LP w.r.t. $w$ and $w'$. Then*

$$|w^\intercal \hat{z} - w'^\intercal \hat{z}'| \le \frac{\max(\|b\|_\infty, |w^\intercal \hat{z}|)}{d} \|w - w'\|_\infty,$$

*where $d$ is the $l_\infty$ distance from $A, b$ to the dual infeasible LP ($\|A, b\|_\infty = \max_{i,j,k}\{|A_{ij}|, |b_k|\}$):*

$$d = \inf\{\delta | \|\Delta A, \Delta b\|_\infty < \delta \Rightarrow \text{the dual problem of the LP with}(A + \Delta A, b + \Delta b) \text{ is infeasible}\}.$$

### 3.2 Approximation Error

In this section, we compare the approximation error of models with different inference algorithms. The discussions are based on the following definition (Definition 1.1 of [Daniely et al., 2012]).

**Definition 8.** For hypothesis spaces $\mathcal{H}, \mathcal{H}'$, we say $\mathcal{H}$ *essentially contains* $\mathcal{H}'$ if for any $h' \in \mathcal{H}'$, there is an $h \in \mathcal{H}$ satisfying $e(h) \le e(h')$ for all $\mathcal{D}$, where $e(h) = \mathbf{E}_\mathcal{D} l(y, h(x))$. In other words, for any distribution $\mathcal{D}$, the approximation error of $\mathcal{H}$ is at most the error of $\mathcal{H}'$.

Our main result is that there exist cases that approximation errors of exact and underestimate inference are not comparable, in the sense that neither $\mathcal{H}$ contains $\mathcal{H}^-$, nor $\mathcal{H}^-$ contains $\mathcal{H}$. [2]

To see that approximation errors could be non-comparable, we consider an approximate inference algorithm $h^-$ which always outputs the second best $y$ for a given $w$. The two examples in Figure 1 and 2 demonstrate that it is both possible that $e(h) < e(h^-)$ and $e(h^-) < e(h)$. The following are the details.

We consider an input space containing two samples $\mathcal{X} = \{x, x'\}$. Sample $x$ has three possible output structures, which are named with $1, 2, 3$ respectively. Sample $x'$ also has three possible $y$, which are named with $1', 2', 3'$. Let the correct output of $x$ and $x'$ be $1$ and $1'$. For sample $x$, feature vectors $\Phi(x, 1), \Phi(x, 2), \Phi(x, 3) \in \mathbf{R}^2$ are points on the unit circle and form a equilateral triangle $\triangle(1, 2, 3)$. Similarly, feature vectors $\Phi(x', 1'), \Phi(x', 2'), \Phi(x', 3')$ are vertices of $\triangle(1', 2', 3')$. The parameter space of $w$ is the unit circle (since inference results only depend on directions of $w$). Given a $w$, the exact inference $h(x, w)$ choose the $y$ whose $\Phi(x, y)$ has the largest projection on $w$ (i.e., $h(x, w) = \arg\max_{y \in \{1,2,3\}} w^\intercal \Phi(x, y)$ and $h(x', w) = \arg\max_{y \in \{1', 2', 3'\}} w^\intercal \Phi(x', y)$), and $h^-(x, w)$ choose the $y$ whose $\Phi(x, y)$ has the second largest projection on $w$.

We first show that it is possible $e(h) < e(h^-)$. In Figure 1, (a) shows that for sample $x$, any $w$ in the gray arc can make the output of exact inference correct (i.e., $h(x, w) = 1$). Similarly, in (b), any $w$ in the gray arc guarantees $h(x', w) = 1'$. (c) shows that the two gray arcs in (a) and (b) are overlapping on the dark arc. For any $w$ in the dark arc, the exact inference has correct outputs on both $x$ and $x'$, which means that approximation error of exact inference $\mathcal{H}$ is 0.

At the same time, in (d) of Figure 1, gray arcs contain $w$ which makes the underestimate inference correct on sample $x$ (i.e., $h^-(x, w) = 1$), gray arcs in (e) are $w$ with $h^-(x', w) = 1'$. (f) shows the gray arcs in (d) and (e) are not overlapping, which means it is impossible to find a $w$ such that $h^-(\cdot, w)$ is correct on both $x$ and $x'$. Thus the approximation error of underestimate inference $\mathcal{H}^-$ is strictly larger than 0, and we have $e(h) < e(h^-)$.

Similarly, in Figure 2, (a), (b), (c) show that we are able to choose $w$ such that the underestimate inference is correct both on $x$ and $x'$, which implies the approximation error of underestimation $\mathcal{H}^-$ equals 0. On the other hand, (d), (e), (f) shows that the approximation error of exact inference $\mathcal{H}$ is strictly larger than 0, and we have $e(h^-) < e(h)$.

Following the two figures, we can illustrate that when $\Phi(x, y)$ are vertices of convex regular n-gons, it is both possible that $e(h) < e(h^-)$ and $e(h^-) < e(h)$, where $h^-$ is an underestimation outputting the $k$-th best $y$. In fact, when we consider the "worst" approximation which outputs $y$ with the smallest score, its approximation error equals to the exact inference since $h(x, w) = h^-(x, -w)$. Thus, we would like to think that the geometry structures of $\Phi(x, y)$ could be complex enough to make both exact and underestimate inference efficient.

To summarize, the examples suggest that underestimation algorithms give us a different family of predictors. For some data distribution, the underestimation family can have a better predictor than the exact inference family.

Finally, for the case of overestimate approximation, we can show that $\mathcal{H}^+$ contains $\mathcal{H}$ using Theorem 1 of [Kulesza and Pereira, 2007].

**Theorem 9.** *For $\rho > 1$, if the loss function $l$ satisfies $l(y_1, y_2) \leq l(y_1, y_3) + l(y_3, y_2)$, then $\mathcal{H}^+$ contains $\mathcal{H}$.*

# 4 Training with the New Margin

Theorems 4 and 6 suggest that we could learn the model parameter $w$ by minimizing a non-convex objective $\mathcal{L}(w, S) + \|w\|^2$. The $\mathcal{L}(w, S)$ term is related to the size of the set $\{y | m_\rho(x_i, y_i^*, y, w) \leq M_i\}$, which can be controlled by margin $m_{\rho^2}(x_i, y_i, y_i^-)$. Specifically, for underestimation,

$$m_\rho(x_i, y_i^*, y, w) \geq \rho w^\mathsf{T} \Phi(x_i, y_i) - w^\mathsf{T} \Phi(x_i, y) \geq \rho w^\mathsf{T} \Phi(x_i, y_i) - w^\mathsf{T} \Phi(x_i, y_i^*)$$
$$\geq \rho w^\mathsf{T} \Phi(x_i, y_i) - \rho^{-1} w^\mathsf{T} \Phi(x_i, y_i^-) = \rho^{-1} m_{\rho^2}(x_i, y_i, y_i^-, w), \quad \forall y.$$

It implies that the larger $m_{\rho^2}(x_i, y_i, y_i^-)$, the lower $\mathcal{L}(w, S)$. Thus, when working with approximate inference, we can apply $m_{\rho^2}(x_i, y_i, y_i^-)$ in existing maximum margin frameworks instead of $m_1(x_i, y_i, y_i^*)$ (replacing exact $y_i^*$ with the approximate $y_i^-$). For example, the structural SVM in [Finley and Joachims, 2008] becomes $\min . \frac{1}{2} \|w\|^2 + C \sum_i \xi_i$, s.t. $m_{\rho^2}(x_i, y_i, y_i^-, w) > 1 - \xi_i$. Intuitively, $m_{\rho^2}$ aims to improve learning process by including more information about inference algorithms. For overestimation, we don't have similar lower bounds as underestimation, but since $m_\rho(x_i, y_i, y_i^+, w) > 1 \Rightarrow m_1(x_i, y_i, y_i^*, w) > \rho^{-1}$, we can apply the margin $m_\rho$ as an approximation of $m_1$.

In practice, since it is hard to obtain $\rho$ for inference algorithms (even it is possible, as $\rho$ must consider the worst case of all possible $x$, a tight $\rho$ maybe inefficient on individual samples), we treat it as an algorithm parameter which can be heuristically determined either by prior knowledge or by tuning on development data. We leave the study of how to estimate $\rho$ systematically for future work.

For empirical evaluation, we examine structural SVM with cutting plane learning algorithm [Finley and Joachims, 2008], and we also adapt two wildly used online structured learning algorithms with $m_\rho$: structured perceptron [Collins, 2002] (Algorithm 3) and online passive aggressive algorithm (PA) [Crammer et al., 2006] (Algorithm 4). The mistake bounds of the two algorithms are similar to bounds with exact inference algorithms (given in the supplementary).

1: $w_0 = \mathbf{0}$
2: **for** $t = 0$ to $T$ **do**
3:     $y_t^{\text{-}} = h^{\text{-}}(x_t, w_t)$
4:     **if** $y_t^{\text{-}} \neq y_t$ **then**
5:         $w_{t+1} = w_t + \rho\Phi(x_t, y_t) - \Phi(x_t, y_t^{\text{-}})$
6:     **end if**
7: **end for**
8: **return** $w = w_T$

Figure 3: Structured perceptron with $m_\rho$.

1: $w_0 = \mathbf{0}$
2: **for** $t = 0$ to $T$ **do**
3:     **if** $m_\rho(x_t, y_t, y_t^{\text{-}}, w) < 1$ **then**
4:         $w_{t+1} = \arg\min_w. \|w - w_t\|_2$
5:         s.t. $m_\rho(x_t, y_t, y_t^{\text{-}}, w) \geq 1$
6:     **end if**
7: **end for**
8: **return** $w = w_T$

Figure 4: Online PA with $m_\rho$.

# 5 Experiments

We present experiments on three natural language processing tasks: multi-class text classification, sequential labelling and dependency parsing. For text classification, we compare with the vanilla structural SVM. For sequential labelling, we consider three tasks (phrase chunking (`chu`), POS tagging (`pos`) and Chinese word segmentation (`cws`)) and the perceptron training. For dependency parsing, we focus on the second order non-projective parser and the PA algorithm. For each task, we focus on underestimate inference algorithms.

## 5.1 Multi-class classification

Multi-class classification is a special case of structured prediction. It has a limited number of class labels and a simple exact inference algorithm (i.e., by enumerating labels). To evaluate the proposed margin constraints, we consider toy approximate algorithms which output the $k$th best class label.

We report results on the 20 newsgroups corpus (18000 documents, 20 classes). The meta data is removed (headers, footers and quotes), and feature vectors are simple tf-idf vectors. We take 20% of the training set as development set for tuning $\rho$ (grid search in [0, 2] with step size 0.05). The implementation is adapted from SVM$^{\text{multiclass}}$ [3].

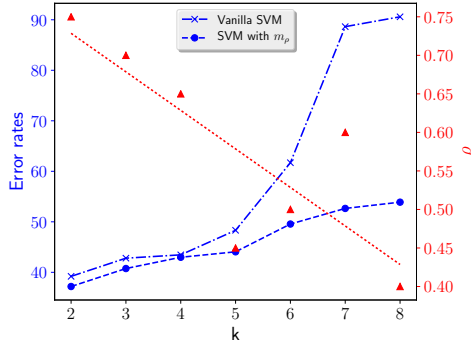

From the results (Figure 5) we find that, comparing with the vanilla structural SVM, the proposed margin constraints are able to improve error rates for different inference algorithms. And, as $k$ becomes larger, the improvement becomes more significant. This property might be attractive since algorithms with loose approximation rates are common in practical use. Another observation is that, as $k$ becomes larger, the best parameter $\rho$ decreases in general. It shows that the tuned parameter can reflect the definition of approximate rate (Defnition 1).

Figure 5: Results on text classification. Blue points are error rates for different $k$, and red points are $\rho$ achieving the best error rates on the development set. The red dot line is the least square linear fitting of red points. The model parameter $C = 10^4$.

## 5.2 Sequential Labelling

In sequential labelling, we predict sequences $y = y^1 y^2, \ldots, y^K$, where $y^k \in Y$ is a label (e.g., POS tag). We consider the first order Markov assumption: $h(x) = \arg\max_y \sum_{k=1}^{K} w^{\mathsf{T}}\Phi(x, y^k, y^{k-1})$. The inference problem is tractable using $O(KY^2)$ dynamic programming (Viterbi).

We examine a simple and fast greedy iterative decoder ("gid"), which is also known as the iterative conditional modes [Besag, 1986]. The algorithm flips each label $y^k$ of $y$ in a greedy way: for fixed $y^{k-1}$ and $y^{k+1}$, it finds a $y^k$ that makes the largest increase of the decoding objective function. The

[3]http://www.cs.cornell.edu/People/tj/svm_light/svm_multiclass.html

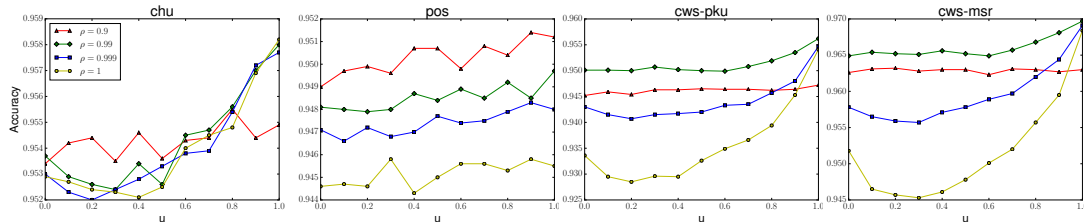

Figure 6: Results of sequential labelling tasks with Algorithm 3. The x-axis represents the random selection parameters $u$. The y-axis represents label accuracy.

algorithm passes the sequence multiple times and stops when no $y^k$ can be changed. It is faster in practice (speedup of 18x on POS tagging, 1.5x on word segmentation), requires less memory ($O(1)$ space complexity), and can obtain a reasonable performance.

We use the CoNLL 2000 dataset [Sang and Buchholz, 2000] for chunking and POS tagging, SIGHAN 2005 bake-off corpus (pku and msr) [Emerson, 2005] for word segmentation. We use Algorithm 3 with 20 iterations and learning step 1. We adopt standard feature sets in all tasks.

To test $\rho$ on more inference algorithms, we will apply a simple random selection strategy to generate a bunch of in-between inference algorithms: when decoding an example, we select "Viterbi" with probability $u$, "gid" with probability $1 - u$. Heuristically, by varying $u$, we obtain inference algorithms with different expected approximation rates.

Figure 6 shows the results of $\rho \leq 1$ [4]. We can have following observations:

- At $u = 0$ (i.e., inference with "gid"), models with $\rho < 1$ are significantly better than $\rho = 1$ on pos and cws ($p < 0.01$ using z-test for proportions). Furthermore, on pos and cws, the best "gid" results with parameters $\rho < 1$ are competitive to the standard perceptron with exact inference (i.e., $\rho = 1, u = 1$). Thus, it is possible for approximate inference to be both fast and good.

- For $0 < u < 1$, we can find that curves of $\rho < 1$ are above the curve of $\rho = 1$ in many cases. The largest gap is $0.2\%$ on chu, $0.6\%$ on pos and $2\%$ on cws. Thus, the learning parameter $\rho$ can also provide performance gains for the combined inference algorithms.

- For $u = 1$ (i.e., using the "Viterbi"), it is interesting to see that in pos, $\rho < 1$ still outperforms $\rho = 1$ by a large margin. We suspect that the $\rho$ parameter might also help the structured perceptron converging to a better solution.

### 5.3 Dependency Parsing

Our third experiment is high order non-projective dependency parsing, for which the exact inference is intractable. We follows the approximate inference in MSTParser [McDonald and Pereira, 2006] [5]. The algorithm first finds the best high order projective tree using a $O(n^3)$ dynamic programming [Eisner, 1996], then heuristically introduces non-projective edges on the projective tree.

We use the online PA in Algorithm 4 with above two-phase approximate inference algorithm. The parser is trained and tested on 5 languages in the CoNLL-2007 shared task [Nivre et al., 2007] with non-projective sentences more than 20%. Features are identical to default MSTParser settings [6].

Figure 1 lists the results with different $\rho$. It shows that on all languages, tuning the parameter helps to improve the parsing accuracy. As a reference, we also include results of the first order models. On Basque and Greek, the performance gains from $\rho$ is comparable to the gains from introducing second order features, but the improvement on Czech, Hungarian and Turkish are limited. We also find that different with text classification and sequential labelling, both $\rho > 1$ and $\rho < 1$ can obtain optimal scores. Thus, with the feature configuration of MSTParser, the value of $w^\mathsf{T}\Phi(x, y^*)$ may not always be positive during the online learning process, and it reflect the fact that feature space of

parsing problems is usually more complex. Finally, setting a global $\rho$ for different training samples could be coarse (so we only get improvement in a small neighborhood of 1), and how to estimate $\rho$ for individual $x$ is an important future work.

| Setting | Basque | Czech | Greek | Hungarian | Turkish |
|---|---|---|---|---|---|
| 1st Order | 79.4 | 82.1 | 81.1 | 79.9 | 85.0 |
| $\rho = 1$ | 79.8 | 82.8 | 81.7 | 81.7 | 85.5 |
| $\rho = 1 - 10^{-3}$ | 79.7 | **83.0** | 81.3 | 81.1 | 85.2 |
| $\rho = 1 - 10^{-4}$ | **80.3** | 82.9 | 82.2 | **81.8** | **85.7** |
| $\rho = 1 + 10^{-3}$ | 79.4 | 82.3 | 81.5 | 80.7 | 85.6 |
| $\rho = 1 + 10^{-4}$ | 79.6 | **83.0** | **82.5** | 81.6 | 85.4 |

Table 1: Results of the second order dependency parsing with parameter $\rho$. We report the unlabelled attachment score (UAS), which is the percentage of words with correct parents.

# 6 Conclusion

We analyzed the learning errors of structured prediction models with approximate inference. For the estimation error, we gave a PAC-Bayes analysis for underestimation and overestimation inference algorithms. For the approximation error, we showed the incomparability between exact and underestimate inference. The experiments on three NLP tasks with the newly proposed learning algorithms showed encouraging performances. In future work, we plan to explore more adaptive methods for estimating approximation rate $\rho$ and combining inference algorithms.

## Acknowledgements

The authors wish to thank all reviewers for their helpful comments and suggestions. The corresponding authors are Man Lan and Shiliang Sun. This research is (partially) supported by NSFC (61402175, 61532011), STCSM (15ZR1410700) and Shanghai Key Laboratory of Trustworthy Computing (07dz22304201604). Yuanbin Wu is supported by a Microsoft Research Asia Collaborative Research Program.

## Footnotes

[1]Definition 1 slightly generalizes "undergenerating" and "overgenerating" in [Finley and Joachims, 2008]. Instead of requiring $\rho > 0$, the "undergenerating" there only considers $\rho \in (0, 1)$, and "overgenerating" only considers $\rho > 1$. Although their definition is more intuitive (i.e., the meaning of "over" and "under" is more clear), it implicitly assumes $w^\mathsf{T} \Phi(x, y^*) > 0$ for all $x$ and $w$, which limits the size of hypothesis space. Finally, by adding a bias term, we could make $w^\mathsf{T} \Phi(x, y^*) + b > 0$ for all $x$, and obtain the same definitions in [Finley and Joachims, 2008].

[2] Note that there exist two paradigms for handling intractability of inference problems. The first one is to develop *approximate inference algorithms for the exact problem*, which is our focus here. Another one is to develop *approximate problems with tractable exact inference algorithms*. For example, in probabilistic graphical models, one can add conditional independent assumptions to get a simplified model with efficient inference algorithms. In the second paradigm, it is clear that approximate models are less expressive than the exact model, thus the approximation error of them are always larger. Our result, however, shows that it is possible to have underestimate inference of the original problem with smaller approximation error.

[4]We also test models with $\rho > 1$, which underperform $\rho < 1$ in general. Details are in the supplementary.

[5]http://sourceforge.net/projects/mstparser/

[6]Features in MSTParser are less powerful than state-of-the-art, but we keep them for an easier implementation and comparison.

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
