[Supplementary Material]

# A Learning Error Analysis for Structured Prediction with Approximate Inference (Supplementary)

**Yuanbin Wu[1,2], Man Lan[1,2], Shiliang Sun[1], Qi Zhang[3], Xuanjing Huang[3]**
[1]School of Computer Science and Software Engineering, East China Normal University
[2]Shanghai Key Laboratory of Multidimensional Information Processing
[3]School of Computer Science, Fudan University
{ybwu, mlan, slsun}@cs.ecnu.edu.cn, {qz, xjhuang}@fudan.edu.cn

## 1 Proofs

### 1.1 Proof of Theorem 4

*Proof.* Let distribution $P, Q$ be both Gaussian:

$$P(w') = \frac{1}{Z_1}\exp\{-\frac{\|w'\|^2}{2}\}, Q(w'|w) = \frac{1}{Z_2}\exp\{-\frac{\|w'-\alpha w\|^2}{2}\},$$

where $\alpha$ is a parameter which will be set later, and $D_{\mathrm{KL}}(Q\|P) = \frac{\alpha^2\|w\|^2}{2}$. We assume the following claim is true and prove it later.

**Claim 1.** *Let $\alpha = (1+\rho)\sqrt{2\ln\frac{2m\lambda_S}{\|w\|^2}}$, then with probability at least $1 - \frac{\|w\|^2}{m}$ over the selection of $w'$, the following holds for any $x_i$:*

$$h^-(x_i, w') \in \{y|m_{\rho_i}(x_i, y_i^*, y, w) \le M_i\} \quad \textit{if } h'(\cdot, w) \in \mathcal{H}^- \tag{1}$$
$$h^+(x_i, w') \in \{y|m_{\rho_i}(x_i, y_i^*, y, w) \ge -M_i\} \quad \textit{if } h'(\cdot, w) \in \mathcal{H}^+. \tag{2}$$

We consider underestimate inference (the proof is similar in the case of overestimate inference). For simplicity, denote the set of $w'$ satisfying Equation (1) by $A_i(w')$. We have

$$\begin{aligned}
&\mathbf{E}_{Q(w'|w)}l(y_i, h^-(x_i, w'))\\
=&\int_{A_i(w')} l(y_i, h^-(x_i, w'))\mathrm{d}Q + \int_{A_i^c(w')} l(y_i, h^-(x_i, w'))\mathrm{d}Q\\
\le&\max_y l(y_i, y)\mathrm{I}(m_{\rho_i}(x_i, y_i^*, y, w) \le M_i) + \frac{\|w\|^2}{m}.
\end{aligned}$$

Summing over $i$ and using Lemma 2, we get the conclusion:

$$\begin{aligned}
&L(Q, S, h(\cdot, w))\\
\le& \frac{1}{m}\sum_{i=1}^m \max_y l(y_i, y)\mathrm{I}(m_{\rho_i}(x_i, y_i^*, y, w) \le M_i) + \frac{\|w\|^2}{m}\\
=& \mathcal{L}(w, S) + \frac{\|w\|^2}{m}.
\end{aligned}$$

We are left to prove the claim. We will first focus on underestimation (Equation 1). From the property of the Gaussian distribution, for any $\varepsilon > 0$ we have

$$\mathbf{P}_{w'\sim Q(w'|w)}(|w'_p - \alpha w_p| \ge \varepsilon) \le 2\exp(-\frac{\varepsilon^2}{2}),$$

where $w_p$ is an element in $w$. For any $y$, using the union bound over all non-zero feature dimension of $\Phi(x_i, y)$, we have $|w'_p - \alpha w_p| < \varepsilon$ with probability at least $1 - 2\lambda_S \exp\left(-\frac{\varepsilon^2}{2}\right)$ over the choice of $w'$. For those $w'$, if some $y$ implies $m_{\rho_i}(x_i, y_i^*, y, w) > M_i$, then

$$
\begin{aligned}
m_{\rho_i}(x_i, h(x_i, w'), y, w') &\geq m_{\rho_i}(x_i, y_i^*, y, w') \\
&= m_{\rho_i}(x_i, y_i^*, y, \alpha w) + m_{\rho_i}(x_i, y_i^*, y, w' - \alpha w) \\
&> \alpha M_i + (w' - \alpha w)^{\mathsf{T}} \Delta_{\rho_i}(x_i, y_i^*, y) \\
&> \alpha M_i - \varepsilon \|\Delta_{\rho_i}(x_i, y_i^*, y)\|_1.
\end{aligned}
$$

Taking $\alpha = \varepsilon(1+\rho)$, we get $m_{\rho_i}(x_i, h(x_i, w'), y, w') > 0$, which means $y \neq h'(x_i, w')$. Finally, the value of $\alpha$ can be obtained by $2\lambda_S \exp\left(-\frac{\varepsilon^2}{2}\right) \leq \frac{\|w\|^2}{m}$. We complete the proof for underestimate approximation.

For overestimation (Equation 2), we have a similar argument. If some $y$ implies $m_{\rho_i}(x_i, y_i^*, y, w) < -M_i$, then

$$
\begin{aligned}
m_{\rho_i}(x_i, h(x_i, w'), y, w') &= m_{\rho_i}(x_i, h(x_i, w'), y, \alpha w) + m_{\rho_i}(x_i, h(x_i, w'), y, w' - \alpha w) \\
&\leq m_{\rho_i}(x_i, y_i^*, y, \alpha w) + m_{\rho_i}(x_i, h(x_i, w'), y, w' - \alpha w) \\
&< -\alpha M_i + (w' - \alpha w)^{\mathsf{T}} \Delta_{\rho_i}(x_i, h(x_i, w'), y) \\
&< -\alpha M_i + \varepsilon \|\Delta_{\rho_i}(x_i, h(x_i, w'), y)\|_1.
\end{aligned}
$$

Taking $\alpha = \varepsilon(1 + \rho)$, we get $m_{\rho_i}(x_i, h(x_i, w'), y, w') < 0$, which means $y \neq h'(x_i, w')$.

$\square$

## 1.2 Proof of Theorem 6

*Proof.* Let $y_i'^* = h(x_i, w'), y_i'^- = h^-(x_i, w'), y_i'^+ = h^+(x_i, w')$, we first establish the following upper bound for $m_{\rho_i}(x_i, y_i'^*, y_i'^-, w')$.

$$
\begin{aligned}
&m_{\rho_i}(x_i, y_i'^*, y_i'^-, w') \\
&= \rho_i w'^{\mathsf{T}} \Phi(x_i, y_i'^*) - w'^{\mathsf{T}} \Phi(x_i, y_i'^-) \\
&\leq \rho_i w'^{\mathsf{T}} \Phi(x_i, y_i'^*) - w'^{\mathsf{T}} \Phi(x_i, y_i'^-) \underbrace{-\rho_i w^{\mathsf{T}} \Phi(x_i, y_i^*) + w^{\mathsf{T}} \Phi(x_i, y_i^-)}_{\geq 0} \\
&= \rho_i w'^{\mathsf{T}} \Phi(x_i, y_i'^*) - \rho_i w^{\mathsf{T}} \Phi(x_i, y_i'^*) + \underbrace{\rho_i w^{\mathsf{T}} \Phi(x_i, y_i'^*) - \rho_i w^{\mathsf{T}} \Phi(x_i, y_i^*)}_{\leq 0} + w^{\mathsf{T}} \Phi(x_i, y_i^-) - w'^{\mathsf{T}} \Phi(x_i, y_i'^-) \\
&\leq \rho_i (w' - w)^{\mathsf{T}} \Phi(x_i, y_i'^*) + \underbrace{w^{\mathsf{T}} \Phi(x_i, y_i^-) - w'^{\mathsf{T}} \Phi(x_i, y_i'^-)}_{\tau\text{-stable}} \\
&\leq \rho_i \|w' - w\|_\infty M_i + \tau \|w' - w\|_\infty M_i \\
&\leq (\rho_i + \tau) \|w' - w\|_\infty M_i.
\end{aligned}
$$

Similarly, for $m_{\rho_i}(x_i, y_i'^*, y_i'^+, w')$, we have the following lower bound.

$$
\begin{aligned}
&m_{\rho_i}(x_i, y_i'^*, y_i'^+, w') \\
&= \rho_i w'^{\mathsf{T}} \Phi(x_i, y_i'^*) - w'^{\mathsf{T}} \Phi(x_i, y_i'^+) \\
&\geq \rho_i w'^{\mathsf{T}} \Phi(x_i, y_i'^*) - w'^{\mathsf{T}} \Phi(x_i, y_i'^+) \underbrace{-\rho_i w^{\mathsf{T}} \Phi(x_i, y_i^*) + w^{\mathsf{T}} \Phi(x_i, y_i^+)}_{\leq 0} \\
&= \rho_i w'^{\mathsf{T}} \Phi(x_i, y_i'^*) - \rho_i w^{\mathsf{T}} \Phi(x_i, y_i^*) + w^{\mathsf{T}} \Phi(x_i, y_i^+) - w'^{\mathsf{T}} \Phi(x_i, y_i'^+) \\
&\geq \rho_i w'^{\mathsf{T}} \Phi(x_i, y_i^*) - \rho_i w^{\mathsf{T}} \Phi(x_i, y_i^*) + \underbrace{w^{\mathsf{T}} \Phi(x_i, y_i^+) - w'^{\mathsf{T}} \Phi(x_i, y_i'^+)}_{\tau\text{-stable}} \\
&\geq \rho_i (w' - w)^{\mathsf{T}} \Phi(x_i, y_i^*) - \tau \|w' - w\|_\infty M_i \\
&\geq -(\rho_i + \tau) \|w' - w\|_\infty M_i.
\end{aligned}
$$

To complete the proof, we are left to establish the following claim which is similar to Claim 1 in Theorem 4.

**Claim 2.** *Let $\alpha = (1+2\rho+\tau)\sqrt{2\ln\frac{2m\lambda_S}{\|w\|^2}}$, then with probability at least $1 - \frac{\|w\|^2}{m}$ over the selection of $w'$, the following holds for any $x_i$:*

$$h^-(x_i, w') \in \{y \mid m_{\rho_i}(x_i, y_i^*, y, w) \leq M_i\}.$$
$$h^+(x_i, w') \in \{y \mid m_{\rho_i}(x_i, y_i^*, y, w) \geq -M_i\}.$$

First, for the case of underestimation, using the property of Gaussian distribution and the union bound (like the proof of Theorem 4), we have $|w'_p - \alpha w_p| < \varepsilon$ with probability at least $1 - 2\lambda_S \exp\left(-\frac{\varepsilon^2}{2}\right)$ over the choice of $w'$. For those $w'$, if some $y$ implies $m_{\rho_i}(x_i, y_i^*, y, w) > M_i$, then

$$
\begin{aligned}
m_{\rho_i}(x_i, h(x_i, w'), y, w') &\geq m_{\rho_i}(x_i, y_i^*, y, w') \\
&= m_{\rho_i}(x_i, y_i^*, y, \alpha w) + m_{\rho_i}(x_i, y_i^*, y, w' - \alpha w) \\
&> \alpha M_i + (w' - \alpha w)^\mathsf{T} \Delta_{\rho_i}(x_i, y_i^*, y) \\
&> \alpha M_i - \varepsilon \|\Delta_{\rho_i}(x_i, y_i^*, y)\|_1.
\end{aligned}
$$

Taking $\alpha = (1 + 2\rho + \tau)\varepsilon$, we get $m_\rho(x_i, h(x_i, w'), y, w') > (\rho + \tau)\varepsilon M_i$, which means $y \neq h(x_i, w')$. Similarly, we can establish the claim for the case of overestimation. $\qquad\square$

## 1.3 Mistake Bounds of Algorithm 3

The mistake bounds results (and their proofs) of structured percetron and PA are almost identical to the results with exact inference algorithms. We only show the bounds of the structured perceptron which is from Collins [2002]. For PA, we refer the readers to Crammer et al. [2006].

**Theorem 3.** *Let $h^-(\cdot, w)$ be a $\rho$-approximation of $h(\cdot, w)$ for all $w$. If there is a $u$ such that $m_\rho(x, y, y^-, u) \geq \delta > 0$ and $R = \sup_{x,y}\|\Phi(x,y)\|_2$, then the number of mistakes made by the structured prediction in Figure 3 is bounded by $\frac{2(1+\rho)R^2}{\delta^2}$.*

*Proof.* Let $N_T$ be the number of mistakes which have been made before round $T$. Following the standard argument,

$$
\begin{aligned}
u^\mathsf{T} w_{t+1} &= u^\mathsf{T} w_t + u^\mathsf{T}(\rho\Phi(x_t, y_t) - \Phi(x_t, y_t^-)) \geq N_T \delta, \\
\|w_{t+1}\|^2 &= \|w_t\|^2 + 2w_t^\mathsf{T}(\rho\Phi(x_t, y_t) - \Phi(x_t, y_t^-)) + \|\rho\Phi(x_t, y_t) - \Phi(x_t, y_t^-)\|^2 \\
&\leq \|v_t\|^2 + 2(1+\rho)R^2 \leq 2N_T(1+\rho)R^2.
\end{aligned}
$$

The inequality is from $w_t^\mathsf{T}\Phi(x_t, y_t^-) > \rho w_t^\mathsf{T}\Phi(x_t, y_t^*)$. $\qquad\square$

*Remark.* Note that the above mistake bound requires $h^-(\cdot, w)$ to be a $\rho$-approximation over all $w$. We could relax the requirement that only for the $u$, $h^-[u]$ is $\rho$-approximation. In order to do so, we need to modify to the update condition in Figure 3 from $y_t \neq y_t^-$ to $\rho w_t^\mathsf{T}\Phi(x_t, y_t) < w_t^\mathsf{T}\Phi(x_t, y_t^-)$. On the other sides, since the $\rho$ is considered to be a tunable parameter in practice, we think that we can use the original update condition without pain.

## 2 Additional Experiments on Sequential Labelling

Figure 1 describes the overall performances over different inference algorithms. Methods for comparison include: the exact inference ("Viterbi"), an algorithm which outputs the second best sequence using dynamic programming similar to the Viterbi ("2nd-bst"), a greedy search algorithm ("greedy"), the proposed greedy iterative decoding algorithm ("gid"), and a mixed algorithm which is a random selection between the Viterbi and greedy iterative decoding with $u = 0.8$ ("mix.8"). We compare the tag accuracy and training time in bar plots, and append F1-values of chunking and word segmentation in tables.

In general, the accuracy decreases as the approximation rate decreases, while the training time will increase. For the greedy iterative decoding, it is faster than the Viterbi on all tasks, especially on

chunking and POS tagging which have large size tag sets (23 and 45). And its accuracy is higher than the "2nd-bst" on chu, pos and msr. Hence, "gid" could be an effective inference algorithm for sequential labelling. For the "mix.8" setting, it makes a clear trade-off between time and accuracy, which shows that the random selection method could be utilized to search for a balanced algorithm.

**chu**

|         | Viterbi | 2nd-bst | greedy | gid   | mix.8 |
|---------|---------|---------|--------|-------|-------|
| Acc     | 0.958   | 0.934   | 0.951  | 0.953 | 0.955 |
| F       | 0.934   | 0.893   | 0.925  | 0.928 | 0.931 |
| Time(s) | 868     | 882     | 57     | 120   | 604   |

**pos**

|         | Viterbi | 2nd-bst | greedy | gid   | mix.8 |
|---------|---------|---------|--------|-------|-------|
| Acc     | 0.945   | 0.938   | 0.940  | 0.945 | 0.945 |
| F       | NA      | NA      | NA     | NA    | NA    |
| Time(s) | 2387    | 2352    | 65     | 136   | 1571  |

**cws-pku**

|         | Viterbi | 2nd-bst | greedy | gid   | mix.8 |
|---------|---------|---------|--------|-------|-------|
| Acc     | 0.954   | 0.946   | 0.925  | 0.934 | 0.939 |
| F       | 0.949   | 0.943   | 0.921  | 0.930 | 0.937 |
| Time(s) | 508     | 605     | 292    | 347   | 400   |

**cws-msr**

|         | Viterbi | 2nd-bst | greedy | gid   | mix.8 |
|---------|---------|---------|--------|-------|-------|
| Acc     | 0.968   | 0.942   | 0.940  | 0.952 | 0.956 |
| F       | 0.967   | 0.953   | 0.943  | 0.953 | 0.959 |
| Time(s) | 1063    | 1306    | 514    | 689   | 738   |

Figure 1: Overview of inference algorithms.

Finally, we give results of random selection experiments with $\rho > 1$ in Figure 2. We can see that the performances are lower than that of $\rho < 1$.

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

Figure 2: Results of the new learning algorithms with $\rho \geq 1$. The x-axis represents the random selection parameters between "gid" and "Viterbi". The y-axis is label accuracy.