[Reviews · NeurIPS 2017]

Reviewer 1



There are some related works with learning-theoretic guarantees that go unmentioned in this manuscript: 1) London et al, "Collective Stability in Structured Prediction: Generalization from One Example", ICML 2013 2) Honorio and Jaakkola, "Structured Prediction: From Gaussian Perturbations to Linear-Time Principled Algorithms", UAI 2016 3) McAllester and Keshet, "Generalization Bounds and Consistency for Latent Structural Probit and Ramp Loss", NIPS 2011 The constant \rho introduced in Definition 1 seems to act as an approximation constant (in theoretical computer science terms.) Recall that in general, inference is NP-hard, and in several instances, it is also NP-hard to approximate within a constant approximation factor (e.g., when Y is the set of directed acyclic graphs.) Authors should comment on this and provide more intuition behind the constant \rho. Finally, given the PAC-Bayes results on McAllester 2007, the proof of Theorem 4 and 6 seem to follow straightforwardly. === After rebuttal I think that the technical part could be improved a bit, by using specific examples (directed trees for instance) and plugging specific worst-case \rho's. All this from the theoretical point of view, in order to better appreciate the results.

Reviewer 2



This is a multi-faceted paper: it contributes new theory, new algorithms, and experiments to evaluate the algorithms. Unfortunately, I do not have the background to evaluate the details of your proofs. However, I found your motivation of the theoretical question good and well-explained. For underestimation you provide an attractive framework for new learning algorithms. Do you have any recommendations for how to leverage the insights of this paper to improve learning with overestimation? The improvements in your experiments are minor, but they help illustrate your point. The main shortcoming of the experiments is that there is no exploration of overestimation. The introduction and abstract would be improved significantly if you defined overestimation and underestimation much earlier on. Perhaps introduce them informally and then define them later with math. For overestimation, I would emphasize early on that this often comes from some LP relaxation of a combinatorial problem.

Reviewer 3



This paper is on the important topic of learning with approximate inference. Previous work, e.g., Kulesza and Pereira (2007), has demonstrated the importance of matching parameter update rules and inference approximation methods. This paper presents a new update rule based on PAC Bayes bounds, which is fairly agnostic to the inference algorithm used -- it assumes a multiplicative error bound on model score and supports both over and under approximations. The example given in section 3.2 is a great illustration of how approximation error is more subtle than we might think it is. Sometimes an approximate predictor can fit the training data better because it represents a different family of functions! The experiments presented in the paper are on NLP problems. However, in NLP the "workhorse" approximate inference algorithm used is beam search. I strongly encourage the authors to extend their experiments to include beam search (varying the beam size to control the approximation quality). If, for some reason, beam search does not match their method, it should be made very clear. Related to beam search is work done by Liang Huang (e.g., http://www.aclweb.org/anthology/N12-1015) about how perceptron with beam search can diverge and how to fix the learning algorithm (see "early update" and "max violation" updates). This paper is pretty influential in NLP. This is a case where the learning rule is modified. I think the NLP community (at least) would be very interested if beam search could actually work with your approach. Overall, I like this paper and I think that it could have a real impact because learning with approximate inference is such an important topic and the approach describe here seems to be pretty effective. Unfortunately, the current state of the paper is a bit too rough for me to recommend acceptance. (If only we had the option to "accept with required revisions") Question ======== I'm confused about Theorem 6. a) Isn't this bound looser than the one in theorem 4? Should making *more* assumptions (about \rho *and* \tau) lead to a tighter bound? b) I expected \rho to be replaced with a bound based solely on \tau. What's the deal? Am I missing something? Lower-level comments ==================== The authors assume that the reader is already familiar with the concepts of under-estimation and over-estimation. I do not believe most readers will know about this. I strongly recommend defining these terms very early in the paper. I would also recommend explaining early on why these two classes of approximations have been studied in the past (and why they are typically studied separately) -- this would be much more useful than the current discussion in the related work section. bottom of page 1: Please define risk? In particular is e(h) /empirical/ risk or true risk? 39: typo? "We investigate three wildly used structured prediction models" -> I think you meant "widely", but I suppose "wildly" also works :P 41: At this point in the paper, I'm very surprised to see "text classification" on a list of things that might use approximate inference... It might be useful to explain how this is relevant -- it seems as if the answer is that it's just a synthetic example. There are some settings, such as "extreme classification", where the number of labels is huge and approximate inference might be used (e.g., based on locality sensitive hashing or approximate nearest neighbors). 42: typo "high order" -> "higher-order" 45: It's tough to say that "Collins [5]" was "the first" since many results for the /unstructured/ multi-label case technically do transfer over. 47,49,50,etc: Citations are not nouns use use \citet instead of \citep and \usepackage{natbib}. "[37] provided..." -> "Collins (2001) provided ..." 49: "Taskar’s bound" is undefined, presumably it's in one of the citations in the paragraph. (Also, you probably mean "Taskar et al.'s bound".) There are many excellent citations in related work (section 2), but without much discussion and clear connection to the approach in the paper. 68: "w the parameter." -> "w is the parameter vector." (probably want to say $w \in \mathbb{R}^d$ as well) 70: clearly over and under estimation method do not partition the space of approximate inference algorithms: what sorts of algorithms does it omit? does it matter? Also, what is the role of \rho? Do we need to know it? does it have constant for all x? (Looks like later, e.g., theorem 4, it's allowed to depend on the example identifier) Why are multiplicative bounds of interest? Which approx methods give such bounds? (Later it is generalized based on stability. Probably worth mentioning earlier in the paper.) The notation h[w](x) is unusual, I'd expect/prefer h_w(x), or even h(w, x). 73, 74: definitions of H, H- and H+ are a little sloppy and verbose. The definition of H is pretty useless. 73: The typesetting of R doesn't mach line 68. Please be consistent. section 3.1 - L(Q,S,h[w]) seems unnecessary (it's fine to just say that the expectation is over S). - Also, S is undefined and the relationship between S and x_i is not explained (I assume x_i \in S and |S|=m). Alternatively E_{x \sim S} would do the trick. Lemma 2: Missing citation to PAC-Bayes Theorem. Please explain the relationship between the prior p(w') and Q(w'|w). The use of a prior comes out of the blue for readers who are unfamiliar with PAC Bayes bounds. A few words about why I should even be thinking about priors at this point would be useful. Also, if I don't care about priors can't I optimize the prior away? 77: Should technically pass Q(.|w) to KL divergence, i.e., $D(Q(.|w) || P(.))$. 80: It might be more useful to simply define under and over estimation with this definition of margin. theorem 4: - assume h'[w] be a ..." -> "theorem 4: assume h'[w] is ..." - Should be a \rho_i approximation of h[w](x_i)? - what is \bar \cal Y ? - why bother subscripting \lambda_S? just use \lambda 90: a better/more-intuitive name for square root term? 107: definition 8, "contains" is not a good term since it's easy to mistaking it for set containment or something like that. Might be better to call it "dominates". Section 3.2 could be summarized as approximate inference algorithms give us a /different/ family of predictors. Sometimes the approximate inference family has a better predictor (for the specific dataset) than the exact family. Giving high-level intuitions such as the one I provided with resonate more with the reader. Figures 1 and 2 take a little too much effort to understand. Consider hoisting some of the discussion from the main text into the caption. At the very least, explain what the visual elements used mean mean (e.g., What is the gray arc? What are the columns and rows?) 159: "... or by tuning." on development data or training data? 246: "non-comparableness" -> "incomparability"